# Improving Prediction of Cervical Cancer Using KNN Imputed SMOTE Features and Multi-Model Ensemble Learning Approach

**DOI:** 10.3390/cancers15174412

**Published:** 2023-09-04

**Authors:** Hanen Karamti, Raed Alharthi, Amira Al Anizi, Reemah M. Alhebshi, Ala’ Abdulmajid Eshmawi, Shtwai Alsubai, Muhammad Umer

**Affiliations:** 1Department of Computer Sciences, College of Computer and Information Sciences, Princess Nourah bint Abdulrahman University, P.O. Box 84428, Riyadh 11671, Saudi Arabia; hmkaramti@pnu.edu.sa; 2Department of Computer Science and Engineering, University of Hafr Al-Batin, Hafar Al-Batin 39524, Saudi Arabia; ralharthi@uhb.edu.sa; 3Department of Computer Science, Faculty of Computing and Information Technology, King Abdulaziz University, Jeddah 21589, Saudi Arabia; ralhebshi@kau.edu.sa; 4Department of Cybersecurity, College of Computer Science and Engineering, University of Jeddah, Jeddah 23218, Saudi Arabia; aaeshmawi@uj.edu.sa; 5Department of Computer Science, College of Computer Engineering and Sciences, Prince Sattam bin Abdulaziz University, P.O. Box 151, Al-Kharj 11942, Saudi Arabia; sa.alsubai@psau.edu.sa; 6Department of Computer Science & Information Technology, The Islamia University of Bahawalpur, Bahawalpur 63100, Pakistan

**Keywords:** cervical cancer detection, missing values, healthcare, KNN imputer, SMOTE, ensemble learning

## Abstract

**Simple Summary:**

This paper presents a cervical cancer detection approach where the KNN Imputer techniques is used to fill the missing values and after that SMOTE upsampled features are utilized to train a multi-model ensemble learning approach. Results demonstrate that use of KNN Imputed SMOTE features yields better results than the original features to classify cancerous and normal patients.

**Abstract:**

**Objective:** Cervical cancer ranks among the top causes of death among females in developing countries. The most important procedures that should be followed to guarantee the minimizing of cervical cancer’s aftereffects are early identification and treatment under the finest medical guidance. One of the best methods to find this sort of malignancy is by looking at a Pap smear image. For automated detection of cervical cancer, the available datasets often have missing values, which can significantly affect the performance of machine learning models. **Methods:** To address these challenges, this study proposes an automated system for predicting cervical cancer that efficiently handles missing values with SMOTE features to achieve high accuracy. The proposed system employs a stacked ensemble voting classifier model that combines three machine learning models, along with KNN Imputer and SMOTE up-sampled features for handling missing values. **Results:** The proposed model achieves 99.99% accuracy, 99.99% precision, 99.99% recall, and 99.99% F1 score when using KNN imputed SMOTE features. The study compares the performance of the proposed model with multiple other machine learning algorithms under four scenarios: with missing values removed, with KNN imputation, with SMOTE features, and with KNN imputed SMOTE features. The study validates the efficacy of the proposed model against existing state-of-the-art approaches. **Conclusions:** This study investigates the issue of missing values and class imbalance in the data collected for cervical cancer detection and might aid medical practitioners in timely detection and providing cervical cancer patients with better care.

## 1. Introduction

Cervical cancer is a form of cancer that arises in the cells of the cervix, the lower region of the uterus that connects to the vagina. Typically, cervical cancer is initiated by an infection resulting from human papillomavirus (HPV), a sexually transmitted infection. HPV is a prevalent virus capable of inducing abnormal alterations in cervical cells, which, if left untreated, can potentially progress into cancer [1].

Cervical cancer is ranked as the third leading cause of death for women, following breast cancer [2] and lung cancer. Unfortunately, it is commonly believed that cervical cancer remains incurable in advanced stages. However, significant progress has been made recently to improve the detection rate of the disease by using imaging techniques. Based on statistics provided by the World Health Organization (WHO), cervical cancer ranks as the fourth most prevalent cancer worldwide. In 2018 alone, around 570,000 new cases were documented, constituting 7.5% of all female cancer-related fatalities [3]. Out of the reported 311,000 annual deaths attributed to cervical cancer, approximately 85% occur in countries with lower- and middle-income economies. Timely detection of cervical cancer plays a crucial role in preserving lives. Women with HIV face a six-fold higher risk of developing cervical cancer compared to those without HIV, and it is estimated that 5% of all cervical cancer cases are associated with HIV. Several factors contribute to the effectiveness of screening, including access to equipment, consistent screening tests, adequate supervision, and the identification and treatment of detected lesions [4].

Cervical cancer can be categorized into two main types: squamous cell cancer, which accounts for 70–80% of cases, and adenocarcinoma, which originates from glandular cells responsible for producing cervical canal mucus. Although squamous cell carcinoma is more common, the occurrence of adenocarcinoma has been on the rise in recent years, now accounting for 10 to 15% of uterine cancers. Detecting adenocarcinoma through screening presents greater challenges as it develops in the cervical canal rather than the cervix itself. However, the treatment approaches for both types of cancer are similar [5,6]. The primary cause of cervical cancer is human papillomavirus (HPV), particularly high-risk types. Several risk factors can increase the likelihood of developing cervical cancer in women infected with HPV. These factors include smoking, early sexual activity, multiple sexual partners, genital herpes infection, a weakened immune system, lower socioeconomic status, poor genital hygiene, and a higher number of childbirths [7,8]. Symptoms of cervical cancer can vary depending on the tumor’s size and the stage of the disease. However, the challenge lies in the pre-cancerous stage, as it often lacks noticeable symptoms and is typically detected incidentally during routine annual check-ups. In advanced stages, approximately 90% of cases present clear symptoms, with irregular vaginal bleeding being the primary symptom associated with cervical cancer.

The process of cervical cancer screening often involves a gynecological examination, which can be painful [9,10] and uncomfortable [11] for patients. The discomfort experienced during the examination can result in delays or avoidance, which hinders early diagnosis. Additionally, inadequate public health policies in developing nations contribute to low rates of cervical cancer screening. As a result, the mortality rate in these countries is 18 times higher [12], with approximately nine out of 10 deaths related to cervical cancer transpiring in low-income countries [13]. Considering that early-stage cervical cancer has relatively high survival rates, reaching up to 90% over a 5-year period [14], it is imperative to improve cervical cancer screening rates. However, screening rates differ between countries, with higher rates observed in developed countries [15] and alarmingly low rates in developing nations.

A range of preventive measures are implemented to combat cervical cancer; however, relying solely on screening tests is insufficient. The timely detection of cervical cancer in its early stages is vital for preventing deaths caused by invasive cervical cancer. Presently, computer vision, machine learning (ML) [16], artificial intelligence (AI) [17], and deep learning (DL) techniques [18] are extensively utilized in disease detection [19]. ML models, in particular, have garnered considerable interest due to their ability to swiftly identify specific diseases [20]. By employing various preprocessing techniques such as data cleaning, dimensionality reduction, and feature selection on the disease dataset, ML algorithms can be applied to achieve precise and accurate results. These analyzed outcomes can aid medical professionals in swiftly diagnosing diseases and providing optimal treatments to patients. This study leverages machine learning techniques for the precise and timely detection of cervical cancer, offering the following key contributions and novelty in the proposed computer-aided diagnosis (CAD) system:A unique ensemble model is put forth in this work to forecast cervical cancer in patients. Extreme gradient boosting (XGB), random forest (RF), and extra tree classifier (ETC) are the foundations of the proposed ensemble model, and a voting mechanism is used to determine the final prediction.The KNN (K nearest neighbor) imputer is used in studies to produce missing values in order to address the issue of missing values.The SMOTE (synthetic minority oversampling technique) is utilized to equalize the class-imbalance problem using the up-sampling technique.The proposed model gives the best accuracy on KNN imputed and SMOTE up-sampled dataset.Different machine learning models, including RF, LR (logistic regression), GBM (gradient boosting machine), GNB (Gaussian Naive Bayes), ETC, SVC (support vector classifier), DT (decision tree), and SGD (stochastic gradient descent), are used to compare their performances. The performance of the suggested model is compared against cutting-edge methods in terms of accuracy, precision, recall, and F1 score in order to assess its efficacy.

The paper is organized as follows: Section 2 offers a detailed analysis of current classification algorithms employed in detecting cervical cancer. In Section 3, the dataset, the proposed methodology for cervical cancer detection utilizing various classification algorithms, and up-sampling techniques are explained. Section 4 primarily focuses on presenting the findings and facilitating discussions. Lastly, Section 5 encompasses the paper’s conclusion and outlines potential avenues for future research.

## 2. Related Work

Machine learning (ML) [21] is an extraordinary tool that finds application in numerous domains, extending to the identification and diagnosis of diseases in diverse animal and plant species. In recent years, numerous ML models have been developed and utilized to enhance research efforts and expedite progress in specific areas of interest. In the context of cervical cancer classification, several studies have been conducted and are discussed in this section of the paper.

Machine and deep learning models are used for different types of medical diagnoses like breast cancer [22], Lung cancer [23], endoscopy [24], and many others. CT images are the most accurate dataset for image-based medical diagnosis [25,26,27]. Some other research works make use of deep learning models for cross-domain work like image-captioning [28,29], drowsy driver detection [30], and neural stem differentiation [31]. CNN applications are also extended to mirror detection with visual chirality cue [32]. In a research study conducted by Kalbhor et al. [33], the discrete cosine transform (DCT) and discrete wavelet transform (DWT) were employed to extract features. To effectively reduce the dimensionality of these features, the fractional coefficient approach was utilized. The reduced features were then utilized as input for seven machine learning classifiers to differentiate between various subgroups of cervical cancer. The study achieved an accuracy of 81.11%. Devi and Thirumurugan [34] conducted another study where they utilized the C-means clustering algorithm to segment cervical cells. Texture features, including the Gray-Level Co-occurrence Matrix (GLCM) and geometrical descriptors, were extracted from these cells. To reduce the dimensionality of the extracted features, principal component analysis (PCA) was employed. Subsequently, the K-nearest neighbors (KNN) algorithm was utilized to classify the cervical cells, resulting in an accuracy of 94.86%.

In their study, Alquran et al. [35] focused on the classification of cervical cancer using the Harvel dataset. They combined deep learning (DL) with a cascading support vector machine (SVM) classifier to achieve accurate results. By integrating these techniques, they successfully classified cervical cancer into seven distinct categories with an impressive accuracy of up to 92%. In their research, Kalbhor et al. [36] introduced an innovative hybrid technique that combined deep learning architectures, machine learning classifiers, and a fuzzy min–max neural network. Their approach focused on the feature extraction and classification of Pap smear images. The researchers utilized pre-trained deep learning models, including AlexNet, ResNet-18, ResNet-50, and GoogleNet. The experimental evaluation was conducted using benchmark datasets, namely, Herlev and Sipakmed. Notably, the highest classification accuracy of 95.33% was achieved by fine-tuning the ResNet-50 architecture, followed by AlexNet, on the Sipakmed dataset.

Tanimu et al. [37] conducted a study focusing on the identification of risk factors associated with cervical cancer using the decision tree (DT) classification algorithm. They utilized recursive feature elimination (RFE) and least absolute shrinkage and selection operator (LASSO) feature selection techniques to identify the most important attributes for predicting cervical cancer. The dataset used in the study had missing values and exhibited a high level of imbalance. To address these challenges, the researchers employed a combination of under and oversampling techniques called SMOTETomek. The results demonstrated that the combination of DT, RFE, and SMOTETomek achieved an impressive accuracy score of 98.72%. Quinlan et al. [38] conducted a comparative analysis to assess different machine learning models for cervical cancer classification. The dataset used in their study exhibited class imbalance, requiring a solution to address this issue. To mitigate the class imbalance problem, the researchers employed the resampling technique called SMOTE-Tomek in combination with a tuned Random Forest algorithm. The results demonstrated that the Random Forest classifier with SMOTE-Tomek achieved a remarkable accuracy score of 99.69%.

Gowri and Saranya [39] proposed a machine learning framework for accurate cervical cancer prediction. Their approach involved the utilization of DBSCAN and SMOTE-Tomek to identify outliers in the dataset. Two prediction scenarios were conducted: DBSCAN + SMOTE-Tomek + RF and DBSCAN + SMOTE + RF. The research findings demonstrated that the DBSCAN + SMOTE + RF approach achieved an impressive accuracy rate of 99%. Abdoh et al. [40] proposed a cervical cancer classification system that utilized the Random Forest (RF) classification technique along with the synthetic minority oversampling technique (SMOTE) and two feature reduction methods: recursive feature elimination and principal component analysis (PCA). The experiment utilized a dataset containing 30 features. The study investigated the impact of varying the number of features and found that using SMOTE with RF and all 30 features resulted in an impressive accuracy of 97.6%.

Ijaz et al. [41] proposed a data-driven system for the early prediction of cervical cancer. Their approach incorporated outlier detection and the SMOTE oversampling method. The classification task was performed using the random forest algorithm in combination with Density-Based Spatial Clustering of Applications with Noise (DBSCAN). Results of their study showed that the DBSCAN + SMOTE-Tomek + RF approach achieved an impressive accuracy score of 97.72% when applied to a dataset with 10 features. Jahan et al. [42] presented an automated system for the detection of invasive cervical cancer. Their research focused on comparing the performance of eight different classification algorithms in identifying the disease. The study involved selecting various top feature sets from the dataset and employed a combination of feature selection techniques, including Chi-square, SelectBest, and Random Forest, to handle missing values. Notably, the MLP algorithm achieved an impressive accuracy of 98.10% when applied to the top 30 features. Mudawi and Alazeb [43] introduced a comprehensive research system consisting of four phases for the prediction of cervical cancer. Their study involved utilizing various machine learning models such as logistic regression (LR), random forests (RF), decision trees (DT), k-nearest neighbors (KNN), Gradient Boosting Classifier (GBC) Adaptive Boosting, support vector machines (SVM), and XGBoost (XGB). The findings revealed that SVM achieved an impressive accuracy score of 99% in the prediction task.

Through an extensive literature survey, it has been observed that various existing approaches have demonstrated favorable performance in predicting cervical cancer across different datasets. Nevertheless, researchers have utilized various optimization techniques to improve performance metrics such as accuracy, precision, and recall. The main aim of this study is to conduct a comparative analysis of different machine learning techniques with the purpose of identifying the most appropriate method for predicting cervical cancer. The complete summary of the related work is shown in Table 1.

## 3. Material and Methods

In this section, we will present a concise introduction to the utilized dataset, the techniques employed for data preprocessing, the machine learning algorithms utilized for detecting cervical cancer, and a summary of the proposed methods for achieving class balance.

### 3.1. Description of the Dataset

For this research, the investigators utilized a dataset obtained from [44], which is publicly accessible and was collected at the Hospital Universitario de Caracas in Venezuela. This particular dataset is currently the only publicly available resource that can be employed for developing a potential survey on cervical cancer screening using AI algorithms and questionnaires. The primary objective of the researchers was to evaluate the feasibility and effectiveness of AI models and class-balancing techniques in analyzing the given dataset for conducting the study.

Table 2 presents a comprehensive summary of the dataset, which consists of a total of 858 instances and 36 attributes. The table provides detailed information on the 35 input variables and one output variable included in the dataset. Each of the input variables is thoroughly described within Table 2.

The dataset contains an output variable called “Biopsy”. Table 2 illustrates that the dataset is characterized by a significant class imbalance. Recognizing the challenges inherent in classifying imbalanced data, the researchers have chosen to address missing values using the kNN imputer technique and tackle the class imbalance issue by employing the SMOTE technique as an up-sampling method.

### 3.2. Data Preprocessing

Data preprocessing plays a crucial role in enhancing the performance of machine learning models. This stage involves eliminating irrelevant or redundant data from the dataset, as such data do not contribute meaningful information for the models. Preprocessing plays a critical role in improving the effectiveness of learning models and also helps in reducing computational time. In this study, while conducting data preprocessing, it was discovered that the dataset contains various missing values. Table 2 displays the distribution of missing values based on the corresponding class. The information presented in Table 2 indicates a significant presence of missing values. Since the dataset consists of categorical data, there are three possible approaches to address these missing values:Employing imputation methods.Eliminating the missing values from the dataset.Removing the missing values and applying the up-sampling technique.

Based on the data preprocessing discussed in this section, it becomes evident that the dataset used for the experiments lacks balance. Specifically, the dataset utilized for cervical cancer prediction is widely employed but suffers from a high-class imbalance. Out of a total of 858 samples, only 58 samples pertain to the cancerous class. This class imbalance poses a risk of model overfitting, as machine learning models tend to assign greater importance to the class with a larger number of samples. Consequently, despite achieving satisfactory accuracy results with the machine learning models, the F1 score is adversely affected. To address this issue, this study proposes the utilization of the SMOTE resampling approach, aiming to enhance the accuracy of cervical cancer detection.

### 3.3. Synthetic Minority Oversampling Technique (SMOTE)

The SMOTE technique is an effective oversampling method commonly employed in medical applications to address the issue of class-imbalanced data [45]. It works by augmenting the number of data instances in the minority class through the generation of synthetic data points from its nearest neighbors using Euclidean distance. These new instances are designed to resemble the original data since they are generated based on the original features. However, it is worth noting that SMOTE may not be the optimal choice for high-dimensional data as it can introduce additional noise. In the context of this study, the SMOTE technique is utilized to generate a new training dataset.

### 3.4. Imputation Methods

#### k-NN Imputer

Throughout the years, various methods have been developed to tackle the issue of missing data and determine replacement values [46]. These methods can be broadly classified into statistical and machine learning approaches. Statistical methods include techniques such as multiple imputation, non-parametric imputation, parametric imputation, and linear regression. On the other hand, machine learning methods involve approaches such as decision tree imputation, neural networks, and k-nearest neighbors (kNN).

The kNN algorithm is commonly employed for imputing missing data by utilizing values from neighboring observations within the same dataset. This technique identifies the k-nearest neighbors of the data point(s) with a missing value(s) and replaces those missing values with the mean or mode value of the corresponding feature values from the k-nearest neighbors. The advantages of kNN imputation include:It does not necessitate constructing a predictive model for each feature containing missing data.k-NN imputer handles both categorical and continuous values.Missing values are efficiently handled by the k-NN imputer.k-NN imputer considers data correlation structure.

It is important to note that kNN is a non-parametric imputation method, which adds to its practicality and flexibility. Missing Values Removal from Dataset.

Another approach for handling the missing values in the data is to simply remove them. In the third set of experiments, this approach is employed, where all the fields containing missing values are eliminated from the dataset.

### 3.5. Supervised Machine Learning Models

This section centers on the discussion of the machine learning algorithms utilized in the study, including their implementation details and hyperparameters. The implementation of these algorithms was carried out using the scikit-learn library and NLTK. A total of eight supervised machine learning algorithms, commonly employed for classification and regression tasks, were utilized in the study. The implementation of these algorithms was done using Python’s scikit-learn module. For the purpose of addressing the classification problem in this study, three specific machine learning algorithms were selected.

#### 3.5.1. Logistic Regression (LR)

Logistic Regression (LR) is a statistical technique employed to analyze data when the goal is to predict an outcome using one or more independent variables [47,48,49]. LR is specifically designed as a regression model that estimates the probability of belonging to a specific category, making it a suitable option for target variables that are categorical. By utilizing a logistic function, LR establishes the connection between the categorical dependent variable and the independent variables, enabling the estimation of probabilities. The logistic function, also known as a logistic curve or sigmoid curve, is characterized by an “S” shape, as depicted in the equation below:(1)$$f\left(x\right)=\frac{L}{1+e^{-m(v-v_{o})}}$$

In the equation provided, the components represent the following:“*e*” represents Euler’s number, the base of the natural logarithm.“*v_o_*” represents the *x*-value of the sigmoid midpoint, indicating the point on the x-axis where the curve reaches its midpoint.“*L*” represents the maximum value or the upper limit of the sigmoid curve.“*m*” represents the steepness of the curve, determining how quickly the curve rises or falls.

#### 3.5.2. Decision Tree

The Decision Tree (DT) is a popular supervised learning method extensively used for solving regression and classification tasks. Its primary goal is to build a predictive model by applying predefined decision rules and advanced analytical techniques to account for prediction errors [50,51]. A decision tree serves as a representation of a segmented estimate and is commonly illustrated using the Sum of Product (SOP) approach. SOP is also known as the Disjunctive Normal Form (DNF). Each branch that originates from the tree’s root and leads to a subtree with the same class corresponds to a specific combination of attributes, while multiple branches converging to the same class indicate a discontinuity. The mathematical representation of entropy (*E*) can be seen in the equation below, where *E* signifies entropy, “*s*” denotes the number of samples, “*Py*” represents the probability of the positive class (yes), “*Pn*” represents the probability of the negative class (no), and “*n*” represents the total number of samples.
(2)E(s)=∑k=0nnk−py×log2Pn

#### 3.5.3. Random Forest

Random Forest (RF) is an ensemble learning algorithm that combines multiple regression and classification trees. Each tree in the forest is trained on a bootstrap sample, and the optimal splitting factors are selected from a randomly chosen sub-set of all features [52,53]. The selection process differs between regression and classification tasks. In regression, the Gini coefficient is used, while variance decrease is employed for classification. For making predictions in both regression and classification, RF calculates either a majority vote or an average. Moreover, the regression method can produce binary outcomes, enabling probabilistic predictions similar to regression analysis. The information gain for a random forest can be calculated using the equation below, where *T* represents the target variable, *X* represents the feature set being split, and *Gain (T, X)* denotes the entropy value after dividing the feature set *X*.
(3)Gain(T,X)=Entropy(T)−Entropy(T,X)

#### 3.5.4. Stochastic Gradient Decent

SGDC (Stochastic Gradient Descent Classifier) operates based on the principles of Logistic Regression (LR) and Support Vector Machine (SVM) [45,54]. It utilizes the convex loss function of LR and serves as a reliable classifier, especially suited for multiclass classification. By employing the one-versus-all (OvA) approach, SGDC combines multiple classifiers. One notable advantage of SGDC is its efficiency in handling large datasets, as it processes a single example per iteration. Due to its regression technique, SGDC is relatively simple to implement and comprehend. However, to achieve optimal results, it is crucial to properly tune the parameters of SGDC. Additionally, SGDC is highly sensitive to feature scaling, underscoring the significance of appropriately scaling the features.

#### 3.5.5. Extra Tree Classifier

ETC (Extra Trees Classifier) is a meta-estimator implementation that improves prediction accuracy by training several weak learners, specifically randomized decision trees, on different sub-sets of the dataset. Similar to the Random Forest (RF), the ETC is an ensemble learning model used for classification tasks [55,56]. The main difference between ETC and RF lies in how the trees within the forest are constructed. ETC builds decision trees using the original training sample, while RF constructs trees based on bootstrap samples obtained from the original dataset. During the creation of each decision tree, at each test node, a random sub-set of k features is provided to the tree. The tree then selects the optimal feature for splitting the data, typically based on a mathematical criterion such as the Gini Index. By utilizing this random feature sub-set, multiple decision trees are generated that are decorrelated from one another.

#### 3.5.6. XGBoost

XGBoost is a high-speed supervised learning algorithm that is employed in this study for accurate and precise water quality classification. One of the key advantages of XGBoost is its regularized learning features, which aid in the refinement of the final weights and mitigate the risk of overfitting [57,58]. The specific algorithm used in this context is as follows:(4)$$\Omega \left(\theta \right)=\sum_{i=1}^{n}d\left(y_{i},{\hat{y}}_{i}\right)+\sum_{k=1}^{k}\beta \left(f_{k}\right)$$
In the given context, the variables can be defined as follows:“*d*” represents the loss function.“*b*” denotes the regularization term.“*y_i_*” represents the predicted value.“*n*” is the number of instances in the training set.“*k*” is the number of trees.

#### 3.5.7. Support Vector Machine

The primary aim of the model is to detect a boundary within a higher-dimensional space, where the datasets are defined by N characteristics. Several hyperplanes can be employed to describe these boundaries, but the objective is to identify the hyperplane with the maximum margin, which corresponds to the greatest distance between data points of different classes. This optimal hyperplane enhances the confidence in accurately classifying future measurements [59,60]. The Support Vector Machine (SVM) method constructs a hyperplane in a high-dimensional or even infinite-dimensional space, enabling various tasks such as data categorization, regression, feature extraction, and filtering. The hyperplane that maximizes the distance to the closest training instances of any category is crucial for achieving optimal performance and leads to a superior solution. This is because a larger margin corresponds to a lower generalization error of the classifier. This principle is described in
(5)$$\left(x_{1},Y-1\right)\cdots \left(x_{n},Y-n\right)$$

In the above equation for n points, where *X* and *Y* denote the class labels, *W* represents the normal vector, and *b* represents the parameter offset of the hyperplane, the definition of a hyperplane is as follows:(6)$$W^{T}\left(x-b\right)=0$$

#### 3.5.8. Gaussian Naive Bayes

In the case of continuous data, it is common to assume that the continuous values associated with each class follow a normal distribution, also known as a Gaussian distribution [61,62]. In such cases, Gaussian Naive Bayes is a suitable algorithm for making predictions based on the characteristics of this normal distribution. The expected probability of a feature is calculated using the equation:(7)$$P\left(x_{i}|c\right)=\frac{1}{\sqrt{2\pi \sigma c^{2}}}exp\left(-\frac{{\left(x_{i}-\mu_{c}\right)}^{2}}{2\sigma c^{2}}\right)$$

In the equation, the symbols represent the following:$x_{i}$: The value or attribute for which the likelihood is being calculated.σ: The standard deviation of the attribute given $x_{i}$.μ: The mean of the attribute given $x_{i}$.

Using these values, the equation calculates the expected probability or likelihood of a feature given a specific value $x_{i}$, taking into account the mean and standard deviation of the attribute.

### 3.6. Proposed Approach for Cervical Cancer Detection

The study utilized a dataset obtained from Kaggle, a reputable source of publicly available datasets. To address missing values and improve the performance of learning models, preprocessing steps were conducted. The KNN imputer was employed to handle missing values. Subsequently, the data was split into a 70:30 ratio, with 70% allocated for model training and 30% for testing.

For cervical cancer detection, the proposed system utilized an ensemble approach called XGB + RF + ETC. Ensemble models are powerful techniques that combine the predictions of multiple models to enhance accuracy and robustness. Each model in the ensemble has its own strengths and weaknesses, and their combination leads to improved overall performance. The proposed approach for cervical cancer detection combines three popular algorithms: XGB, RF, and ETC. The workflow diagram of the proposed approach is depicted in Figure 1.

The ensemble model operates by combining the predictions generated by three distinct machine learning algorithms. The general methodology for constructing an ensemble model involves training multiple models on the same dataset and subsequently merging their predictions. In the case of the XGB + RF + ETC ensemble model, this methodology is followed by training XGB, RF, and ETC models separately on the identical dataset. Each of these models produces predicted probabilities for each class of the target variable. These predicted probabilities can then be aggregated to generate a final prediction for each observation in the dataset. A common approach to combining the predictions is by calculating a weighted average of the predicted probabilities, with the weights determined based on each model’s performance on a validation set.

The proposed ensemble model operates by leveraging the strengths of three distinct machine learning algorithms to generate predictions that are both accurate and robust. By training multiple models on the cervical cancer dataset and merging their predictions, we can enhance the model’s ability to generalize and mitigate overfitting. The proposed ensemble model functions can be summarized as follows: (8)$$\hat{p}=argmax\left\{\sum_{i}^{n}XGB_{i},\sum_{i}^{n}RF_{i},\sum_{i}^{n}ETC_{i}\right\}.$$
where $\sum_{i}^{n}XGB_{i}$, $\sum_{i}^{n}RF_{i}$, and $\sum_{i}^{n}ETC_{i}$ represent the prediction probabilities for each test sample generated by the *XGB*, *RF*, and *ETC* models, respectively. Subsequently, the probabilities for each test case obtained from *XGB*, *RF*, and *ETC* are passed through the soft voting criterion, as depicted in Figure of proposed voting Figure 2.

The ensemble model selects the final class by considering the highest average probability among the classes and combining the predicted probabilities from both classifiers. The ultimate prediction is determined based on the class with the highest probability score, as
(9)VC(XGB+RF+ETC)=argmax(g(x))

### 3.7. Evaluation Parameter

To evaluate the efficiency of the suggested CAD (Computer-Aided Diagnosis) system, four indices are computed: True Negative (*TN*), True Positive (*TP*), False Negative (*FN*), and False Positive (*FP*). These indices provide information about the correct and incorrect recognition of examples as either positive or negative. Using these indices, several evaluation metrics can be calculated to further analyze the performance of the CAD system. Commonly used evaluation metrics include sensitivity, specificity, accuracy, *F*1-score, precision, and recall. These metrics offer insights into various aspects of the system’s performance. The calculation of these evaluation metrics can be described using the following equation:(10)$$Accuracy\left(A\right)=\frac{TP+TN}{TP+TN+FP+FN}$$
(11)$$Precision\left(P\right)=\frac{TP}{TP+FP}$$
(12)$$Recall\left(R\right)=\frac{TP}{TP+FN}$$
(13)$$F1\text{-}Score\left(F\right)=2\times \frac{Precision\times Recall}{Precision+Recall}$$

## 4. Experiments and Analysis

This section presents the experimental results and discusses their implications, focusing on evaluating the effectiveness of the proposed method in comparison to existing approaches. The evaluation encompasses multiple practical test parameters applied to the cervical cancer dataset, and these results are compared against other machine learning (ML) methods. For conducting the experiments, a Dell PowerEdge T430 machine with 2 GB of RAM is used for training. The machine is equipped with a graphical processing unit (GPU) and runs on a 2× Intel Xeon processor with eight cores and a clock speed of 2.4 GHz. Additionally, it has 32 GB of DDR4 RAM. These specifications provide the necessary computational resources to perform the experiments and evaluate the proposed method’s performance against other ML approaches.

### 4.1. Results of the Machine Learning Model with Deleted Missing Values

The first stage of the experiments consisted of addressing the presence of missing values within the dataset. Subsequently, the modified dataset was subjected to machine learning models. The results obtained from the machine learning models after eliminating the missing values from the dataset are displayed in Table 3.

The results reveal that the RF, ETC, and XGBoost classifiers demonstrated the highest accuracy rates, achieving 71.55%, 72.98%, and 73.41% respectively. RF exhibited a precision of 79.25%, a recall of 80.65%, and an F1 score of 80.11%. ETC showcased a precision of 80.25%, a recall of 80.25%, and an F1 score of 80.25%. Similarly, XGBoost achieved a precision of 79.85%, a recall of 79.99%, and an F1 score of 79.91%. In contrast, LR performed the least effectively, with an accuracy rate of 63.47%, a precision of 76.44%, a recall of 78.54%, and an F1 score of 77.41%.

The proposed VC (XGB + RF + ETC) ensemble system demonstrated superior performance compared to all other learning models, achieving an accuracy rate of 79.93%, a precision of 83.36%, a recall of 85.21%, and an F1 score of 84.67%. However, when considering the individual machine learning models using the dataset without missing values, their performance was unsatisfactory overall.

### 4.2. Results of Machine Learning Models by Using KNN Imputer

In the following phase of the experiments, the KNN imputer was utilized to address the missing values within the dataset. Upon preprocessing the data, it was noticed that certain values were absent, necessitating the application of the KNN imputer to fill in these gaps. The imputation process involved employing the mean of the available values and the Euclidean distance metric. Subsequently, the modified dataset was employed to train and assess various machine learning models. The performance of different models is detailed in Table 4.

The findings indicate that RF, ETC, and XGBoost achieved accuracy scores of 81.65%, 83.10%, and 83.52%, respectively. The proposed VC (XGB + RF + ETC) ensemble model outperformed them all with an accuracy rate of 95.39%. Moreover, the proposed ensemble model demonstrated a precision value of 97.63%, a recall value of 95.96%, and an F1 score of 96.76%. In contrast, the linear model LR had the lowest accuracy value of 73.57%.

### 4.3. Results of Machine Learning Models by Using SMOTE

In the third round of experiments, the SMOTE technique was applied to tackle the issue of class imbalance in the dataset. During data preprocessing, it was noticed that out of the total 858 samples, only 58 samples belonged to the cancerous class. To address this class imbalance problem, SMOTE was used as an oversampling technique. The modified dataset was then employed to train and evaluate multiple machine learning models. Table 5 presents the performance of various models.

The results emphasize that the proposed voting ensemble model VC (XGB + RF + ETC) outperforms all other models, achieving an impressive accuracy of 94.24%. Similarly, the XGB, RF, and ETC classifiers also achieved respectable accuracy scores of 85.37%, 83.48%, and 84.19%, respectively. The tree-based ensemble model DT attained an accuracy value of 78.64%. Among all models, the regression-based model LR and the probability-based model GNN exhibited the lowest accuracy values of 75.47% and 72.34%, respectively. However, the ensemble of linear models VC (XGB + RF + ETC) demonstrates superior performance on the up-sampled dataset.

### 4.4. Results of Machine Learning Models Using KNN Imputed Dataset and SMOTE

The outcomes of the fourth set of experiments, which employed the KNN imputer to handle missing values and SMOTE to address the class imbalance, are presented in Table 6. The combined use of KNN imputer and SMOTE aims to tackle both missing values and class imbalance simultaneously, with the expectation of enhancing the accuracy of the linear model. Machine learning models were trained and evaluated following the application of KNN imputer and SMOTE.

### 4.5. Comparison of Machine Learning Model Results

To assess the effectiveness of the KNN imputer and SMOTE, we conducted a comparison of the machine learning models’ performance in four different scenarios: (i) using KNN imputer, (ii) without KNN imputer, (iii) using SMOTE for up-sampling, and (iv) using both KNN imputer and SMOTE. The comparison revealed that in the fourth experiment, where the KNN imputer was employed first and then SMOTE was used for up-sampling, the performance of the machine learning models showed a significant improvement compared to the results obtained in the previous three experiments. To provide clarity and facilitate performance analysis, Table 7 presents the outcomes of the machine learning models for all scenarios.

Figure 3 illustrates a comparison of the performance of different machine learning models using the complete set of experiments. The graph clearly indicates that incorporating the KNN imputer in conjunction with SMOTE (Synthetic Minority Oversampling Technique) enhances the performance of individual models, resulting in an overall improved performance across all machine learning models.

### 4.6. Performance Comparison with Existing Studies

To assess the efficacy of the proposed approach, a comparative analysis of its performance is conducted against state-of-the-art models that specifically focus on cervical cancer detection. This evaluation involves considering a selection of recent studies from the literature, which serve as benchmarks for comparison. In [37], a cancer detection model utilizing RFE and DT with SMOTETomek achieves an accuracy of 98.82%, precision of 87.5%, recall of 100%, and F1 score of 93.33%. Another study, ref. [38], attains a 99.69% accuracy by employing an up-sampling technique. In [40], PCA features combined with SMOTE + RF yield a 97.6% accuracy, while study [41] employs 10 features for the same task and achieves a 97.72% accuracy score of 97.23% precision, 97.42% recall, and 97.72% F1 score. Additionally, refs. [42,43] report accuracy scores of 98.10% and 99% respectively. Despite the high accuracy reported in the mentioned research works, the proposed models exhibit superior results, as evidenced by Table 8.

The reasons behind the superior performance of the proposed method compared to existing approaches lie in two factors; handling missing values and ensemble voting classifier. The unique combination of techniques, addressing missing values, ensemble learning, and class imbalance handling, are the key factors contributing to the observed improvements in accuracy. Unlike some of the previous methods that may not explicitly address the issue of missing values, this study incorporated a KNN imputation technique coupled with SMOTE up-sampled features. Furthermore, the proposed method employs a stacked ensemble voting classifier that integrates the predictions of three individual classifiers. This ensemble approach often proves beneficial by reducing overfitting, leveraging the strengths of multiple classifiers, and providing more robust predictions.

### 4.7. Results of K-Fold Cross-Validation

K-fold cross-validation is also performed to verify the performance of the proposed model. Cross-validation aims at validating the results from the proposed model and verifying its robustness. Cross-validation is performed to analyze whether the model performs well on all the sub-sets of the data. This study makes use of five-fold cross-validation and results are given in Table 9. Cross-validation results reveal that the proposed ensemble model provides an average accuracy score of 0.996 while the average scores for precision, recall, and F1 are 0.998, 0.998, and 0.997, respectively.

## 5. Conclusions

In recent years, cervical cancer has been considered the leading cause of premature mortality among women. The developing countries cover the major portion (almost 85%) of this deadliest disease according to WHO report. An early diagnosis and timely treatment could greatly help to reduce the fatality rate of cervical cancer. In this regard, the use of machine learning approaches is found to provide higher detection accuracy. This research work proposed a framework that consists of two portions for accurately diagnosing cervical cancer in patients. The first step is to normalize the dataset by using the KNN-imputed SMOTE features and the second part consists of the usage of the stacked ensemble voting classifier (XGB + RF + ETC) model. The results with a high accuracy of 99.99% reveal that the use of ensemble models can provide a reliable solution for the early detection of cervical cancer. The comparison with other state-of-the-art models also shows the superiority of the proposed model. The future work of this research work is to make a stacked ensembling of machine and deep learning models to further enhance the performance of the model on higher dimension datasets and provide generalized and robust results.

## Figures and Tables

**Figure 1 cancers-15-04412-f001:**
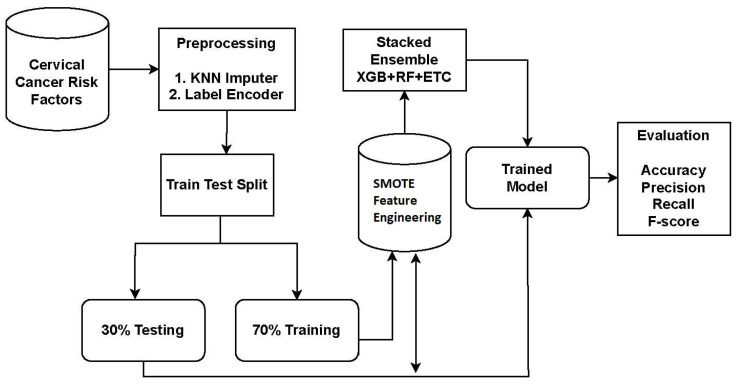
Workflow diagram of the proposed methodology.

**Figure 2 cancers-15-04412-f002:**
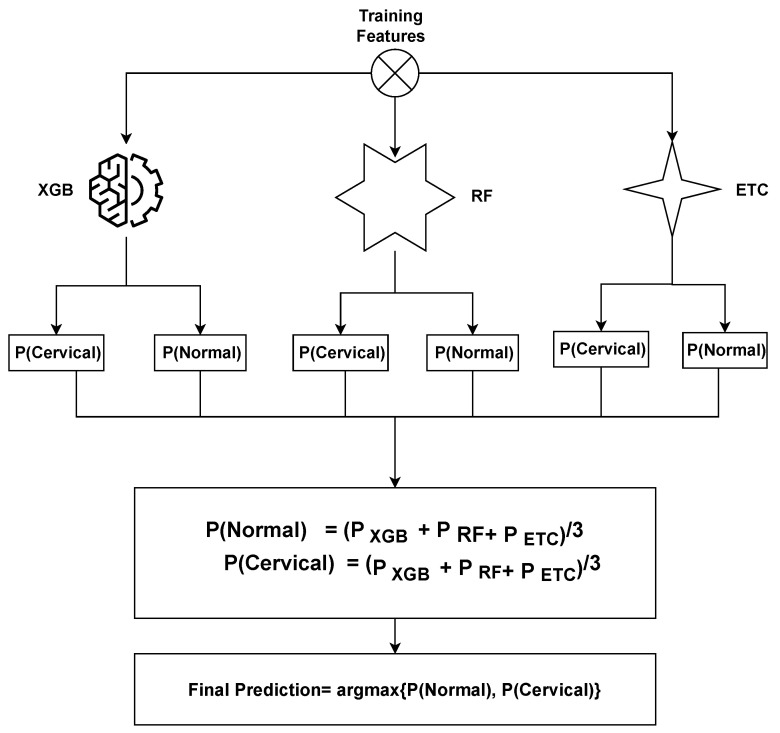
Architecture of the proposed voting classifier.

**Figure 3 cancers-15-04412-f003:**
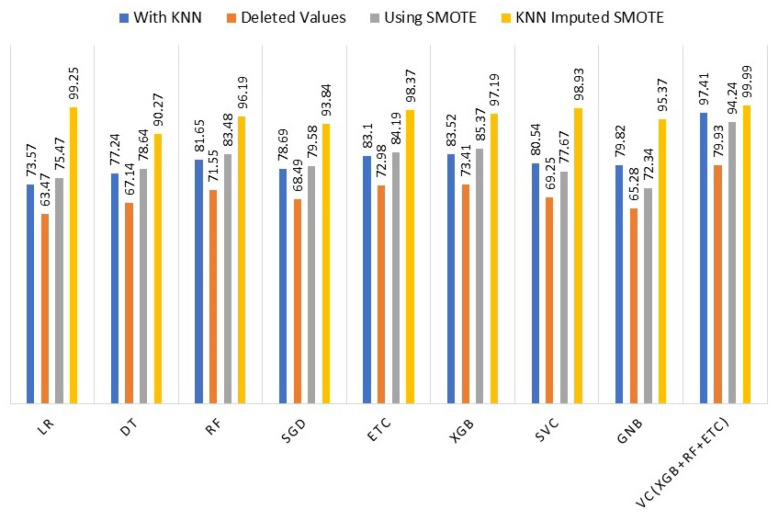
Accuracy Comparison of the machine learning models using all techniques.

**Table 1 cancers-15-04412-t001:** Summary of the related work.

Ref	Classifier	Dataset	Achieved Accuracy
[33]	Simple logistic, Random forest, Navie Bayes, BayeNet, Part, Random Tree, Decision table,	Herlev dataset	81.11% random forest with DCT transformer
[34]	k-NN, Linear Discriminant, Ensemble Bagged trees, and Gaussian SVM	Pap smear image dataset	94.15% KNN with PCA
[35]	CNN, ResNet101, with cascading support vector machine	Herlev dataset	92% deep learning
[36]	Alexnet, ResNet-18, ResNet-50, and GoogleNet	Herlev and Sipakmed	95.33% ResNet-50
[37]	DT + LASSO + SMOTETomek, DT + RFE + SMOTETomek	Same	98.72% DT + RFE + SMOTETomek
[38]	RF, k-NN, KM, DT-E, DT-G, SVM, GB, GNB, LDA, LR	Same	99.69%. Random forest with SMOTETomek
[39]	RF, DBSCAN + SMOTE + RF, DBSCAN + SMOTETomek + RF	Same	99.007% DBSCAN + SMOTE + RF
[40]	SMOTE, RF with different number of PCA features	Same	97.6% using SMOTE with RF and 30 features
[41]	iForest + SMOTE + RF, DBSCAN + SMOTETomek + RF, iForest + SMOTETomek + RF, and DBSCAN + SMOTE+ RF	Same	97.72% DBSCAN + SMOTETomek + RF, on 10 features
[42]	MLP, k-NN, GBC, LR, DT, SVC, ADA, and RF	Same	98.10% MLP on top 30 features
[43]	DT, SVM, RF, LR, KNN, XGB, Adaptive boosting, GBC	Same	99% SVM

**Table 2 cancers-15-04412-t002:** Dataset description.

Number	Attribute Name	Type	Range	Missing Values	% of Missing Values
1	Age	int	13–84	0	0%
2	IUD (Years)	int	0–19	117	13.6%
3	STDs: genital herpes	bool	0–1	105	12.2%
4	Harmonal contraceptives	bool	0–1	108	12.5%
5	Dx: cancer	Bool	0–1	0	0%
6	Smokes	Bool	0–1	13	1.5%
7	STDs: vaginal condylomatosis	Bool	0–1	105	12.2%
8	STDs: AIDS	Bool	0–1	105	12.2%
9	Num of Pregnancies	Int	0–110	56	6.5%
10	Intrauterine Device (IUD)	Bool	0–1	117	13.6%
11	STDs: cervical condylomatosis	Bool	0–1	105	12.2%
12	STDs: molluscum contagiosum	Bool	0–1	105	12.2%
13	STDs: time since last diagnosis	Int	0–3	787	91.7%
14	Cytology	Bool	0–1	0	0%
15	First sex intercourse(age)	Int	10–32	7	0.08%
16	Hormonal contraceptives (years)	Int	0–22	108	12.5%
17	STDs: condylomatosis	Bool	0–1	105	12.2%
18	STDs: Time since first diagnosis	Int	0–1	787	91.7%
19	Schiller	Bool	0–1	0	0%
20	Number of sexual partners	Int	1–28	26	2.6%
21	Smokes (packs/year)	int	0–37	13	1.5%
22	STDs (number)	Int	0–4	105	12.2%
23	STDs: pelvic inflammatory diease	Bool	0–1	105	12.2%
24	STDs: Number of diagnosis	Int	0–1	0	0%
25	Hinselmann	Bool	0–1	0	0%
26	Diagnosis: Dx	Bool	0–1	0	0%
27	STDs: Hepatitis B	Bool	0–1	105	12.2%
28	Smokes (years)	int	0–37	13	1.5%
29	Sexually Transmitted Disease (STD)	Bool	0–1	105	12.2%
30	STDs: syphilis	Bool	0–1	105	12.2%
31	Dx: Human Papillomavirus (HPV)	Bool	0–1	0	0%
32	STDs: vulvo-perineal condylomatosis	Bool	0–1	105	12.2%
33	STDs: HPV	Bool	0–1	105	12.2%
34	Dx: cervical intraepithelial Neoplasia (CIN)	Bool	0–1	0	0%
35	STDs: HIV	Bool	0–1	105	12.2%
36	Biopsy (target Variable)	bool	0–1		

**Table 3 cancers-15-04412-t003:** Results of the machine learning models obtained by deleting missing values from the dataset.

Model	A	P	R	F
LR	63.47	76.44	78.54	77.41
DT	67.14	77.41	79.35	78.67
RF	71.55	79.25	80.65	80.11
SGD	68.49	76.27	78.78	77.56
ETC	72.98	80.25	80.25	80.25
XGB	73.41	79.85	79.99	79.91
SVC	69.25	76.24	81.34	78.52
GNB	65.28	74.34	75.02	74.89
VC (XGB + RF + ETC)	79.93	83.36	85.21	84.67

**Table 4 cancers-15-04412-t004:** Results of the learning models using KNN imputer.

Model	A	P	R	F
LR	73.57	86.54	88.64	87.51
DT	77.24	87.51	89.45	88.77
RF	81.65	89.35	90.88	90.31
SGD	78.69	86.41	88.83	87.86
ETC	83.10	90.33	90.33	90.33
XGB	83.52	89.74	90.25	90.01
SVC	80.54	88.42	89.43	89.25
GNB	79.82	86.43	86.20	86.98
VC (XGB + RF + ETC)	97.41	97.63	95.96	96.76

**Table 5 cancers-15-04412-t005:** Results of the learning models using SMOTE on Original Dataset.

Model	A	P	R	F
LR	75.47	79.34	80.24	79.92
DT	78.64	80.15	81.19	80.96
RF	83.48	85.34	86.34	86.00
SGD	79.58	82.54	83.27	83.08
ETC	84.19	87.24	88.92	88.24
XGB	85.37	87.85	88.68	88.31
SVM	77.67	78.34	79.74	78.37
GNN	72.34	74.71	75.58	75.27
VC (XGB + RF + ETC)	94.24	94.89	95.19	95.06

**Table 6 cancers-15-04412-t006:** Results of Machine Learning Models Using KNN Imputed Dataset and SMOTE.

Model	A	P	R	F
LR	95.58	99.18	98.25	99.25
DT	98.37	90.14	91.96	90.27
RF	93.55	95.19	96.49	96.19
SGD	98.89	92.22	93.19	93.84
ETC	97.91	97.38	98.35	98.37
XGB	95.68	97.67	98.88	97.19
SVM	97.79	98.49	99.49	98.93
GNN	92.88	94.32	95.39	95.37
VC (XGB + RF + ETC)	99.99	99.99	99.99	99.99

**Table 7 cancers-15-04412-t007:** Accuracy Comparison of the machine learning models using all techniques.

Model	With KNN	Without KNN	Using SMOTE	Using KNN Imputer + SMOTE
LR	73.57	63.47	75.47	99.25
DT	77.24	67.14	78.64	90.27
RF	81.65	71.55	83.48	96.19
SGD	78.69	68.49	79.58	93.84
ETC	83.10	72.98	84.19	98.37
XGB	83.52	73.41	85.37	97.19
SVM	80.54	69.25	77.67	98.93
GNB	79.82	65.28	72.34	95.37
VC (XGB + RF + ETC)	97.41	79.93	94.24	99.99

**Table 8 cancers-15-04412-t008:** Comparison with state-of-the-art approaches.

Ref	Approach	A	P	R	F
[37]	DT + RFE + SMOTETomek	98.82%	87.51%	100%	93.33%
[38]	RF + SMOTETomek	99.69%	-	-	-
[39]	DBSCAN + SMOTE + RF	99.07%	-	-	-
[40]	SMOTE + RF	97.60%	98.48%	96.65%	-
[41]	DBSCAN + SMOTETomek + RF	97.72%	97.23%	97.42%	97.72%
[42]	MLP	98.10%	98%	98%	98%
[43]	SVM	99.00%	99.5%	96%	98%
**Proposed**	**VC (XGB + RF + ETC)**	**99.99%**	**99.99%**	**99.99%**	**99.99%**

**Table 9 cancers-15-04412-t009:** Results for k-fold cross-validation of the proposed ensemble model.

Fold Number	Accuracy	Precision	Recall	F-Score
Fold-1	99.23	99.96	99.94	99.95
Fold-2	99.34	99.96	99.95	99.96
Fold-3	99.45	99.97	99.96	99.96
Fold-4	99.11	99.94	100.0	99.99
Fold-5	99.24	99.99	99.98	99.99
**Average**	**99.27**	**99.96**	**99.96**	**99.97**

## Data Availability

The datasets can be found by the authors at request.

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
