# Peer review of "Improving Prediction of Cervical Cancer Using KNN Imputed SMOTE Features and Multi-Model Ensemble Learning Approach"

_cancers, 2023, doi:10.3390/cancers15174412_

Round 1

Reviewer 1 Report

In this manuscript, the authors demonstrate that the use of machine learning approaches provides higher detection accuracy. Considering that cervical cancer is considered to be the leading cause of premature mortality among women and early diagnosis and prompt treatment could go a long way in reducing the mortality rate of this disease, the current research is interesting and helpful. Some grammatical and stylistic errors need to be corrected but the conclusions are in line with the introduction and the methodology is adequate.

no

Author Response

The response is submitted in a separate PDF file.

Reviewer 2 Report

The author(s) did a good job with this manuscript. The overall presentation, background study, methods, results, and discussion are quite good. I had a few comments and suggestions for the authors that should be addressed before a final decision:

1. The author(s) applied different ML classifiers for training (70%) and testing (30%) the proposed dataset. They did not use any validation sets to optimize the parameters of those classifiers. How they had chosen the best parameter that performed in both training and testing.

2. In Tables 3–6, results were shown using ACC, precision, recall, and f1-score measurements, and the efficacy was compared with other state-of-the-art models that were only based on ACC. Why not in terms of precision, recall, and f1? ACC is not the best proof of efficacy for medical disease diagnosis.

3. Table 8 shows that the proposed method outperformed others in terms of ACC. The author(s) should discuss more about why this proposed method outperformed others and what the novelty was that arose here. Because in refs [20] to [26], they also used the same classifier for the same data.

4. In lines 76–79, authors must provide a few references that are utilized in DL, AI, ML, or CNN/DL+ML for disease detection. I'm suggesting a few to add there: https://doi.org/10.1016/j.future.2019.12.033; "Development of Automated Diagnostic Tools for Pneumoconiosis Detection from Chest X-Ray Radiographs. The Final Report Prepared for Coal Services Health and Safety Trust(2019)";https://doi.org/10.3390/ijerph182413409;https://doi.org/10.1145/3378936.3378968;https://doi.org/10.3322/caac.21552;

5. On Line 84, the authors talked about key contributions and novelty with the proposed method. ML classification,  SMOTE for imbalance data, and their application are not unique or novel things. Hence, it was also applied with the same dataset already. How will they address the novelty issue of this proposed method?

6. For this manuscript, any graphical depiction of the results of the model testing will be more striking, and this will highlight the overall calibre and its viability.

There are a few mistakes. Please revise thoroughly and make changes as required.

Author Response

(The authors gave the same response as above.)

Round 2

Reviewer 2 Report

No additional comments are required. Its quality has been enhanced overall.

There are some grammatical and aesthetic mistakes that need correction.